# Suitability of the Attitudes to Aging Questionnaire Short Form for Use among Adults in Their 50s: A Cross-Sectional e-Survey Study

**DOI:** 10.3390/ijerph20227035

**Published:** 2023-11-08

**Authors:** Gail Low, Alex Bacadini França, Donna M. Wilson, Gloria Gutman, Sofia von Humboldt

**Affiliations:** 1Faculty of Nursing, University of Alberta, Edmonton, AB T6G 1C9, Canada; dmwilson@ualberta.ca; 2Laboratory of Human Development and Cognition, Federal University of São Carlos, São Paulo 13566-590, Brazil; alex.tonante@gmail.com; 3Gerontology Research Centre, Simon Fraser University, Vancouver, BC V6B 5K3, Canada; gutman@sfu.ca; 4William James Center for Research, ISPA—Instituto Universitário, 1149-041 Lisbon, Portugal; sofia.humboldt@gmail.com

**Keywords:** attitudes to aging, midlife, psychometrics, survey, Canada

## Abstract

This cross-sectional e-survey study examines the suitability (reliability and validity) of the 12-item Attitudes to Aging Questionnaire Short Form (AAQ-SF) for use among adults in their 50s. The AAQ-SF instrument was originally designed to capture subjective perceptions of physical change, psychosocial loss, and psychological growth by asking people aged 60 and beyond how they feel about growing older. Our sample comprised 517 people residing in three Canadian provinces. Respondents completed the Attitudes to Aging Questionnaire Short Form, the Rosenberg Self-Esteem Scale, and a short sociodemographic profile. Our findings replicate the original AAQ-SF structure for physical change, psychosocial loss, and psychological growth, with a promising internal consistency range for the third subscale. In our sample, psychological growth is best represented as ‘Self’ and ‘Generativity’, with a particularly greater capacity to explain variations in scores for item 18 and item 21. Physical change and psychosocial loss scores strongly differed based on perceived health and chronic illness presence. Psychosocial loss and psychological growth were moderately correlated with two aspects of self-esteem. We relate these patterns of findings within the context of prevailing growth and development theory and their perceived implications within the context of COVID-19 and post-pandemic life.

## 1. Introduction

The Attitudes to Aging Questionnaire Short Form (AAQ-SF) is a twelve-item, three-subscale tool developed through a secondary analysis of World Health Organization Quality of Life Group data collected across 20 countries [1]. The parent AAQ-24 measures self-perceived age-related physical changes (PCs), psychosocial losses (PLs), and psychological growth (PG) [2]. These 24 items were generated through focus groups with people 60+ years of age, all of whom were considered experts about their position in life [3]. Respondents are, therefore, asked to think about how they feel about growing older and their life as a whole when completing the AAQ-24 [4]. Because items are not pre-weighted [2], no assumptions can be made about their importance. Notably, positive and negative aspects of aging are both considered necessary to provide a balanced personal perspective on aging [5]. All AAQ-24 study centers followed the WHO’s translation methods and reviews by same-language and bilingual researchers and practitioners to ensure semantic equivalency [6]. 

Along with its rich cross-cultural heritage, a completion time of less than 5 min for the AAQ-SF is expected to entice time- and energy-constrained participants. In keeping with the pedigree of the AAQ-24 [4,7,8,9], thus far, the lay-friendly AAQ-SF shows promise. The AAQ-SF has exhibited internal consistency reliabilities of α = 0.62 for PG and α = 0.72 for PL and PC. A superior goodness-of-fit of the AAQ-SF’s three-factor versus single-factor structure and no less superior fit than a three-factor structure loading onto a fourth higher-order factor was observed among an at-random split-half 20-country sample of some 2400 respondents and further confirmed among a third independent studied sample (*n* = 792) [1]. The PG subscale seemingly features two different types of growth, one pertaining to passing on the benefits of one’s experiences and setting a good example for younger people, and the other to the pleasantries and privileges of aging itself. These growth-related items moderately correlated with the AAQ-24 PG subscale as a whole [1]. The AAQ-SF has exhibited strong discriminant abilities for perceived health, depression, anxiety, and quality of life [1]. The less burdensome AAQ-SF appears to be a reliable and valid tool for use across various cultures from age 60 onward. Our study’s aim was to determine its suitability (reliability and validity) for use among people in their 50s.

In midlife, people’s functional abilities have peaked [10]. At the same time, they are at ‘the crossroads of youth and old age’ [11] (p. 20). These crossroads, however, are marked by shifts away from gains toward an increased risk for age-related functional losses [12,13]. Age-related changes in functioning cause people to question who they are and serve as cues that they are losing their youth and that life is finite [14]. Performing as or looking ‘old’ typically carries negative connotations [15]. Across 45 countries, for example, poorer physical health [16] and functioning abilities [17] were linked with negativity toward aging and older people from age 50 onwards. It is noteworthy, however, that people in their 50s and early 60s can be more negative than those in their 70s and 80s about their physical prowess and the pace at which they are aging [18].

Marital dissolution, job loss, losses of loved ones, and financial constraints also begin to surface around age 45 and increase in prevalence well into people’s 50s [19]. Losses in social status in their 40s can have lingering effects on positive or negative affect [20] and can take a cumulative toll on perceived worthwhileness in life for people in their 50s [21]. Expected life events can conjure in midlife adults an inconsolable sense of urgency [11]. Perhaps this is why age-related negativity in people in their 50s darkens their perceived quality of life and social relationships and, perhaps, even provokes long-term physical health neglect [16] and dissatisfaction with work and leisure activities [22]. Age-related morale, including the sense of usefulness, appears to decline to a lesser degree in the 40s versus the 60s [23], but this is not so in the 50s [24]. 

The fifth decade of life is often a time of unrest characterized by life stressors that should, in theory, be of no surprise, but these can dampen how people see themselves, aging itself, and their mental well-being. With people now having to adjust to and develop healthy post-pandemic lives and livelihoods while COVID-19 lingers on, this position in the life course warrants empirical attention within and across countries.

## 2. Materials and Methods

### 2.1. Study Design and Participants

In this cross-sectional e-survey study, data were collected in July and August 2019 by Zoomer Media, an affiliate of the Canadian Association of Retired Persons, or CARP. Canadians can become CARP members as early as age 45 (average age = 61 years of age) and can become Zoomer subscribers as early as age 35 (average age = 57 years of age) [25,26]. Other characteristics among CARP members taking part in the national polls the organization conducted are as follows: 55% were males; 74% were married/common law; they had an average household income of $130,000; and 77% lived in households with two to five people or more [25]. The characteristics of Zoomer subscribers are as follows: 44% were males, with an average age of 57 and an average household income of $78,556 [26]. About 61% of CARP members have a post-secondary degree [25], as well as about 54% of Zoomer subscribers [27]. Marital status and number of household members data are not available for Zoomers.

### 2.2. Eligibility Criteria

Eligibility criteria for this study were as follows: (1) being 50 to 59 years of age; (2) having provided CARP/Zoomer magazine with an email address to receive communications like surveys; and (5) freely consenting to participate in this survey.

### 2.3. Data Collection Procedure

An e-blast (Appendix A) about the study aim, team, and a survey hyperlink was sent to CARP members and Zoomer subscribers in the three provinces in which most lived: Ontario, British Columbia, and Alberta [25,26]. To ensure full anonymity, the names, IP addresses, and, therefore, the number of persons who initially received the digital advertisement were not shared with any study investigator. The study involved a voluntary e-survey. No incentive, monetary or non-monetary, was offered. Recipients who clicked on the survey hyperlink were first taken to a study information letter and an informed consent form, which asked the following: ‘I agree to participate in the research study described: yes/no’. Among the 779 survey hyperlink users tallied by Survey Monkey® [28], 517 (66.37%) were ‘yes’ respondents who also went on to complete all study questionnaires. Survey Monkey’s® [28] advanced survey logic prevented respondents from taking our survey more than once. All study data were encrypted and stored on a password-protected computer in the primary investigator’s department.

#### Measures

Attitudes to Aging Questionnaire Short Form (AAQ-SF, [1]). On this 12-item, 3-subscale instrument, there are 4 items within the PC, PL, and PG subscales. Self-report ratings for items 1 to 3 range from 1 to 5, where 1 reflects strongly disagree, 2 reflects disagree, 3 reflects uncertain, 4 reflects agree, and 5 reflects strongly agree. Response categories for items 4 to 12 are as follows: 1 = not at all true; 2 = slightly true; 3 = moderately true; 4 = very true; and 5 = extremely true. The 4 negatively worded PL items (item 3: ‘Old age is a depressing time of life’; item 5: ‘I see old age mainly as a time of loss’; item 7: ‘As I get older, I find it difficult to make new friends’; item 10: ‘I feel excluded from things because of my age’) are reverse coded so that higher scores reflect more positive attitudes. Subscale scores are derived by adding scores for all 4 items and, therefore, range from 4 to 20.

Health Status. Health status was measured by asking the following: ‘How is your health these days?’. Response categories were poor, fair, good, very good, and excellent. The number of chronic illness response categories originally ranged from 0 to 5 or more. As very few respondents reported having 4 (0.6%) and no respondents reported having 5 or more chronic illnesses, frequency counts were collapsed into 0, 1, 2, and 3 or more chronic conditions.

Rosenberg’s Self-Esteem Scale (RSES, [29]). Rosenberg’s ten-item Self-Esteem Scale measures perceptions of the self in general across two subscales. Self-confidence items include the following: ‘I am satisfied with myself’ and ‘I am a person of worth, at least on an equal plane with others’. Self-deprecation items include the following: ‘I do not have too much to be proud of’ and ‘I certainly feel useless at times’. All 5 items in each subscale are measured on a 4-point Likert scale, ranging from strongly agree to strongly disagree. For self-confidence, a score of 3 is assigned to a ‘Strongly Agree’ response. In the self-deprecation subscale, ‘Strongly Agree’ renders a score of 0. Higher scores indicate higher self-esteem. Our study’s reliability estimates for self-competence (ω = 0.87 [95% CI = 0.85–0.89]; CR = 0.85) and self-deprecation (ω = 0.87 [95% CI = 0.86–0.89]; CR = 0.873) were robust [30,31,32].

### 2.4. Statistical Analyses

All basic statistical analyses were performed with SPSS^®^ 27 (IBM Corp., Armonk, NY, USA) and R^®^ Statistical Software 4.2.3 [33] using the Lavaan [34] and semTools packages [35]. The reliability of the AAQ-SF was also assessed using McDonald’s Omega (ω; cut-off criterion = 0.70) [30,31] and composite reliability (CR; cut-off criterion = 0.70) [31,32], with the latter accounting for measurement error.

A categorical (ordinal) confirmatory factor analysis (CFA) of the 3-subscale, 12-item AAQ-SF structure [1] was conducted using a weighted least squares mean and variance adjusted (WLSMV) estimation method based on polychoric correlations due to the categorical nature of item responses [36,37]. When items are treated as ordinal, all patterns of response across our studied sample of midlife adults are used to estimate item-to-domain loading [38], with our cut-off criterion being 0.50 [39]. As a collective whole, these patterns were not normally distributed with respect to skewness (Mardia’s coefficient = 1087.70; *p* < 0.05) and kurtosis (Mardia’s coefficient = 21.12; *p* < 0.05) [40]. The WLSMV estimation method is recommended for ordinal data that are not normally distributed [41,42]. This estimation method fits with the measurement philosophy of the AAQ-24 and AAQ-SF developers [3].

Model Chi-Square (X2) statistics with *p*-values > 0.05 are ideal; however, true-population models are prone to dismissal with samples of *n* ≥ 250 [43]. Adjunct goodness-of-fit assessments were as follows: (1) comparative fit index (CFI) and (2) Tucker–Lewis index (TLI). Goodness-of-fit could be attributed to models exhibiting CFI and TLI values 0.94 or higher, with sample sizes > *n* = 250 and models espousing 12 observed variables or survey items [44]. The third assessment statistic is the root mean square error of approximation (RMSEA), with values of less than 0.07 [44] and a 90% confidence interval upper bound of up to 0.08 being reasonable approximations [40]. The RMSEA and CFI are far less prone to inflation [43,45,46] and are, therefore, recommended in validation studies [47]. All such indexes are sensitive to incorrectly specified item-to-domain structures with sample sizes greater than *n* = 250 [45]. These indexes were reported in the international study [1]. The latter said ranges for goodness-of-fit assessments are more practical; researchers cannot judge the merits of their models using any ‘single magic value’ (p. 640, [44]). A model’s validity should also be judged using theoretical and practical criteria [40,46,48,49].

Independent t-tests permitted assessments of discriminant validity. In the international study of older persons, all AAQ-SF subscales strongly differed based on perceived health [1]. In AAQ-24 studies, biological sex [50,51,52] and education level [7,51,52] mattered little to adults in their late 40s and 50s. However, being married enhances PG appraisals, and higher numbers of chronic illnesses dampen PC appraisals [51]. Our study is the first to examine differences based on sex, education, and marital status.

We assessed the convergent validity [49] of the AAQ-SF based on its correlations with the RSES. For this, we used structural equation modeling with a WLSMV estimation [36,37]. As a collective whole, RSES scores were not normally distributed (*p* < 0.05) with respect to skewness (Mardia’s coefficient = 1526.23) or kurtosis (Mardia’s coefficient = 47.51) [40]. We tested convergent validity in a full model with the PG, PL, and PC domains (each with 4 items) and the self-deprecation and self-confidence subscales (each with 5 items). Our cut-off criterion for AAQ-SF domain and RSES subscale correlations was r = 0.40 or higher [53], with more positive attitudes toward aging being associated with higher self-regard. Because self-deprecation items are negatively oriented items (i.e., ‘I certainly feel useless at times’), we expected PL (i.e., ‘I feel excluded from things because of my age’) to most strongly correlate with this subscale. Self-confidence should, therefore, most strongly correlate with PG.

A minimum sample size was calculated using Soper’s online SEM sample size calculator [49,54]. Given our item-loading cut-off criterion [39], the AAQ-SF structure (3 latent variables and 12 observed variables), and a desired power of 80% and α of 0.05, our required sample size was *n* = 100 [54]. For the convergent validity analysis, given our cut-off criterion [53] for observed associations between the AAQ-SF and RSES (2 latent variables and 10 observed variables), it was *n* = 88 [54]. 

## 3. Results

Study respondents (*n* = 517) were in our target age range of 50–59 years and, on average, 56.28 years of age (SD = 2.82 years of age). Just over three-quarters of respondents were female. Respondents’ level of education was as follows: 12.4% reported the highest level of education (Master’s or Ph.D.), followed by 32.6% who had higher levels of degrees (Bachelor’s), and 55% had a medium level of education (college or high school degree). In terms of marital status, 10.8% were single, 23.06% were widowed, separated, or divorced, and 66.14% were married or partnered. The vast majority of respondents (94.2%) self-identified as heterosexual. Over half (52.4%) of respondents reported having no chronic illnesses; others were living with one (32.3%), two (9.7%), and three or more (5.5%) chronic illnesses. Few respondents felt that they were in poor or fair (9.86%) or excellent (18.56%) health; most rated their health as good (30.17%) or very good (41.39%).

Respondents felt more optimistic about age-related psychosocial loss (CR = 0.72) than physical change(Table 1). Internal consistency reliability was highest for the PC subscale (CR = 0.78), lowest for the PG subscale (CR = 0.61), and robust for the scale as a whole (ω = 0.80 [0.78–0.83]; CR = 0.88). All item–subscale correlations were strong (*p* < 0.001). The average score for self-deprecation was M = 10.70 (SD = 3.56), and for self-confidence, it was M = 11.08 (SD = 3.06). The total RSES score was, on average, 21.78 (SD = 6.19).

In our categorical CFA of the proposed AAQ-SF structure [1], goodness-of-fit was evaluated using the CFI, the TLI, and the RMSEA, and its 90% confidence interval [40,44,46] (Table 2). With the CFI and TLI (>0.94), the results offer partial evidence of goodness-of-fit (*X*^2^ = 159.89; df = 51; *p* < 0.001; CFI = 0.980; TLI = 0.974). So too does the RMSEA statistic of 0.039, with its 90% confidence interval upper bound (0.025–0.052) being well below 0.08.

As shown in Figure 1a, item loadings were moderate (λ = 0.38 for PG item 18 ‘pass on the benefits of my experience’) to strong (λ = 0.85 for PC item 14 ‘more energy now than I expected’). The loading coefficients for item 18 and item 21 (λ = 0.47) per se did not meet our cut-off criterion. Item-explained variance was lowest for item 18 (14.5%) and item 21 (22.3%) (versus 33.7% for item 2 and 47.7% for item 5). These findings support Laidlaw et al.’s [1] suspicion that the PG subscale represents two distinct types of growth, one akin to generativity or passing on the benefits of one’s experiences and setting a good example for others, and the other to the pleasantries and privileges of aging itself.

Using the same criteria (Table 2), we also assessed an alternative structure (Figure 1b) representing PG as ‘Self’ (item 2 ‘privilege to grow old’ and item 5 ‘pleasant things about growing older’) and ‘Generativity’ (item 18 ‘pass on the benefits of my experience’ and item 21 ‘want to give a good example’). Our sample size was sufficient for testing a four-latent variable structure [54]. For this alternative two-sub-domain PG structure (X^2^ = 116.14, df = 48, *p* < 0.001; X^2^ difference = 43.75, df difference = 3, *p* = 0.001; CFI = 0.927, TLI = 0.900; RMSEA = 0.05 [90% CI = 0.04–0.065]), the CFI and TLI fall below our desired range [44]. However, loading coefficients for item 18 (λ = 0.58) and item 21 (λ = 0.76) were much stronger. The explained variance was also greater for item 18 (34%) and item 21 (59.3%) compared with 36.4% for item 2 and 55.1% for item 5.

Each AAQ-SF subscale was most strongly differentiated on the basis of perceived health (Table 3). Scores for PC and PL were statistically significantly higher among respondents living with chronic illnesses. Our application of the Bonferroni correction [55] rendered PL and PG scores differences between married or partnered versus separated, widowed, divorced, or single respondents not statistically significant. This was also the case for PC scores among those with post-secondary (Master’s or PhD, Baccalaureate, College) versus those without post-secondary education (high school/other). Men’s scores across all subscales were generally not statistically significantly higher than women’s.

Our assessment of convergent validity [49,53] was conducted by examining correlations between the three AAQ-SF domains [1] and the two RSES subscales [29]. In our convergence model (*X*^2^ = 454.12; *df* = 205; *p* < 0.001; *CFI* = 0.866; *TLI* = 0.849; *RMSEA* = 0.049 [*90% CI =* 0.043–0.055]) (Table 4), the CFI and TLI fell below our desired range [44]. All of the correlations shown in Figure 2 are positive and highly statistically significant (*p* < 0.001). Item loadings for the RSES (λ = 0.65 to λ = 0.83) reflect our robust reliability estimates for this instrument. As expected, the PL domain was most strongly correlated with self-deprecation (λ = 0.51 versus λ = 0.42 for self-confidence). The PG domain was also most strongly correlated with self-confidence (λ = 0.47 versus λ = 0.29 for self-deprecation). Respondents’ PC domain scores had borderline moderate correlations with their self-confidence (λ = 0.35) and self-deprecation (λ = 0.34) scores.

## 4. Discussion

This study attempted to determine the suitability (reliability and validity) of the AAQ-SF for use among people in their 50s. Internal consistency reliabilities for all subscales were replicates of those reported in the AAQ-SF study of adults aged 60–100 across 20 countries [1]. Our more robust CFI, TLI, and RMSEA estimates are likely owing to our ordinal-level CFA. All patterns of response among our studied sample of midlife adults were used to estimate item-to-domain loadings. These loadings closely approximate those generated using maximum likelihood estimation (MLE), even when patterns of response are not normally distributed [38]. Our confirmatory item loadings within our PC subscale were strikingly similar to the MLE loadings reported in the international study of older adults. Physical changes most strongly pertained to having more energy and being in better health than expected, followed by keeping fit by exercising and a feeling of being ‘old’. These changes resonated in a similar way with AAQ-SF respondents in their 50s.

Item loadings in the PL subscale were moderate to strong, with item 17 (‘more difficult to make new friends’) having the lowest loading and item 6 (‘old age is a depressing time of life’) having the highest loading. In the international study [1], item loadings ranged from 0.565 for ‘feel excluded from things because of my age’ (item 22) to 0.693 for ‘old age is mainly a time of loss’ (item 12). These patterns hint that what constitutes the least and most poignant age-related losses may differ in the age 50 group compared to later in life. It will be interesting to see whether our PL item-loading patterns are replicated in other studies of midlife adults in Canada and abroad. It is noteworthy that these findings fit with guiding theories particular to the population under study [46,48,49].

Our paler loading (λ = 0.50 versus λ = 0.643 in the international study) for item 17 or ‘difficulty making new friends’ could be rooted in developmental differences. Midlife adults tend to invest more time in satisfying work-related and family-versus-friends’ needs [56]. In keeping with their fewer remaining life years, later-life adults purposefully most strongly invest in meaningful friendship ties [57,58,59]. Some attribute this difference to amplified reciprocal exchanges of support that help older people better cope with their own health declines and significant others passing away [60]. In theory, friendships take on heightened importance and resonate more with people in later life.

Our findings also suggest that the true reliability of the PG AAQ-SF subscale likely ranges between ω = 0.57 and ω = 0.68 in the general population of Canadians in their 50s. In the international AAQ-SF study [1], weak affiliations for item 2 ‘privilege to grow old’ (0.227) and item 5 ‘pleasant things about growing older’ (0.191) were indicators that PG captures two qualitatively different attitudes toward growth. In our case, we found strong loadings for items 2 and 5 and moderate loadings for items 18 and 21. In other words, PG primarily represented growth in relation to the self (items 2 and 5) and, thereafter, generativity (item 18 ‘pass on the benefits of my experiences’ and item 21 ‘want to give a good example’). In addition to having theoretical goodness-of-fit, these differences should be further evaluated as competing models tested against the same data [46,48,49]. Our hierarchy of PG loadings fits with the notion that midlife adults tend to be well-invested in but can be torn between their own and others’ betterment [10,11,19], hence our testing of an alternative ‘Self’ and ‘Generativity’ subscale structure for PG.

Fit indices alone tell us that Laidlaw et al.’s [1] initial model with the four-item PG subscale best represents our sample data [39,48]. Goodness-of-fit indices are, however, one yardstick with which to judge the validity of models [40,46,49]. There are theoretical and empirical reasons for further exploring the merits of PG for self-growth and generativity. Midlife adults are at the crossroads of youth and old age [11], navigating between adapting to age-related changes in functioning and investing in the well-being of younger and older generations through mentoring and caregiving [10]. Negative life events like physical health problems, divorce, and losses of paid work and loved ones begin to surface in people’s 40s and increase in their 50s [19]. Perhaps this is why self-compassion is so pivotal to positive mental health among people in their 40s and 50s [61]. There also appears to be a remarkable growth in wisdom about life’s uncertainties, including death and serious illness, between the ages of 51 and 99 [62]. It is important to focus on the ‘why’ of goodness-of-fit, such as through qualitative explorations of midlife adults’ perceived meanings and relevance of all four PG items [63]. Qualifying (as opposed to just quantifying) observed austerities in goodness-of-fit can help us better understand why items behave as they do across models and potential others to consider.

As was the case in the international study [1], all subscale scores were significantly higher among our respondents with favorably perceived health. Living with chronic illnesses was associated with far greater negativity toward PC and PL, with similar patterns being observed among older Spaniards [5], and for PC alone among middle-aged Australians and Canadians [51].

We found negligible sex-based differences across all AAQ-SF subscales. A lack of differences has been observed across all AAQ-24 subscales among middle-aged Australian, Canadian, and Chinese men and women [50,51,52]. Post-secondary-educated respondents in this study were generally more positive about the physical changes of aging. Being married/partnered was advantageous for PG and PL appraisals. Similar growth-related sentiments have been expressed among midlife adults in Australia and Canada [51]. However, the latter two non-health-related differences were not statistically significant when comparing the number of comparison groups.

In our convergence model, the AAQ-SF subscales behaved as expected in relation to self-confidence and self-deprecation [40,46,49]. There are good theoretical and empirical reasons why the positive and significant correlations between the AAQ-SF and RSES subscales are evidence of the convergent validity of the AAQ-SF. Unlike PG, the PC domain had borderline moderate correlations with the two RSES subscales. People in their 50s could be more inclined to self-identify with the psychosocial versus physical aspects of aging. In keeping with others’ midlife adult findings [64], our respondents were most positive about growth opportunities in older age. The sense of growth is said to play an important role in protecting mental well-being in late midlife [19], with partially fulfilled personal goals and aspirations providing promise of a brighter future [10,11]. Growth-related optimism has strongly differentiated between living a very good and a very poor quality of life from age 60 onward [1]. Happiness and psychological well-being also appear to remarkably improve into the sixth decade of life [65,66].

We are also cognizant that people’s preferences for positive emotional states can increase with age [57,58,59]. For example, older people with higher life satisfaction scores have been shown to be more likely to endorse self-confidence items on the RSES [67,68]. In a recent study of midlife adults, those tending to be more confident about self-managing chronic illness had higher scores on positively worded items on the Taiwanese AAQ-SF [69]. Among the midlife adults in our study, PG scores were highest overall and most strongly correlated with self-confidence. The debate over whether the RSES best fits one construct with positively worded items or two (positively and negatively worded) remains unresolved [70,71]. Choices regarding dimensionality are based on other fundamental considerations [40,46,49]. Some advice is that if a theory or previous empirical findings suggest that positively and negatively oriented items measure distinct constructs, treat them accordingly [70]. Our two-subscale approach should also have practical appeal. An instrument’s ability to capture multiple aspects of people’s lives has long been considered most helpful for policymakers, practitioners, and the populations with whom they work [72]. How people see their own process of aging can impact their mental health in their 50s and well into later life [16]. Knowing whether opportunities for PG are largely tied to self-confidence, not self-deprecation, is therefore important, particularly in post-pandemic times.

The findings of the present study have implications. The COVID-19 pandemic is a historic global life event fraught with unexpected life-threatening illness and deaths, social disconnectedness, and lost livelihoods, all of which may result in mental harm [73,74]. A national online survey of Canadians 55+ years of age revealed that more in the 55 to 64 age group than those 65 and over experienced disruptions in everyday life routines ranging from work to hobbies, changes in employment, and stress and discord in the family [75]. Additionally, midlife adults were more likely than older adults to rate their overall health as worse and to experience negative emotions like sadness and a feeling of being judged. Other research foresaw heightening workloads as putting people at risk for mental harm [76]. In a study of largely middle-aged adults, relational conflicts, difficulty accessing everyday resources, health, time-constrained caregiving, and family separation were significantly associated with symptoms of depression [77]. People in their 50s and early 60s were the second most prominent age group needing emergency work income relief in the early stages of COVID-19 [78]. Monetary losses were subsequently linked with heightened anxiety [79] and mental distress [80] and were, as is the case with forgone exercise [81], exacerbating symptoms of depression [77].

In a similar vein, negative life events can make people increasingly age-aware [12] and more prone to reacting to everyday stressors with significant emotional negativity [82]. The COVID-19 pandemic likely magnified the physical and psychosocial stresses and strains of midlife. Such turmoil has been characterized as a form of suffering [83] but could, on the other hand, be a provocateur of growth that builds mental resilience [76]. Either way, the road to recovery is expected to be long [84], and how people think about aging may never be the same [85,86]. In Germany, some early-to-late middle-aged adults have expressed significant over-time dissatisfaction with their social ties and lives in general [87]. Own-health negativity procured similar sentiments over age-related losses and gains among middle-aged Italians [88]. Physical symptoms of COVID-19 that often disrupt home, work, and school life can also linger for more than one year [89].

Along with the loss of previously secure, familiar social spaces like workspaces, another frightening consequence of large-scale traumatic events is the threat to one’s physical safety, particularly when personally relevant [90]. At the crossroads of youth and old age, such losses and threats would be particularly compelling. In this global recovery phase, cross-cultural studies are needed to better understand how linkages between perceptions of aging and mental health have changed following COVID-19, especially after social distancing [91]. In year one of this pandemic, there were unprecedented increases in the prevalence of anxiety and depression [92,93], perhaps signifying a fallout pandemic [84,94]. On the other hand, little is known about perceived opportunities for psychological growth among midlife adults within the context of COVID-19. Negative COVID-related emotions like fear and sadness are perceived by some in their 70s and 80s as enhancing self-growth [95].

Researchers wanting to investigate both negative and positive aspects of aging should find our pre-COVID findings useful points of comparison and a source of questions to consider. For example, within the context of the COVID-19 pandemic and now, post-pandemic, would opportunities for PG still largely pertain to growth for the self versus others’ betterment? Having weathered arguably exacerbated developmental stresses and strains, would PG be more related to resilience than self-deprecation? Some midlife adults expressed growing negativity toward their lives in general in the first year of the pandemic [87]. Others have sat at crossroads between more anxious younger and less anxious older counterparts in the first two years of the pandemic [96]. Might post-pandemic sentiments be the same? We also wondered whether making friends might carry more weight within this context, particularly for women. Women reported experiencing greater mental harm with social distancing and lockdowns [75,97]. Perhaps then we also ought to ask: would energy and health expectations resonate more with middle-aged or older persons? Practitioners and researchers who aim to assess the mental health-related impacts of the COVID-19 pandemic on middle-aged adults should consider the AAQ-SF as an assessment tool and our data where they lack pre-COVID test scores. It should also be considered for use as a before and after measure to quantify the effects of mental health-enhancing programs on middle-aged adults.

## 5. Limitations and Conclusions

The present study should be interpreted with the following limitations in mind: Our study was conducted in only three out of ten provinces in a single country among a single 10-year age group. Due to a lack of published studies that used AAQ-SF, we also largely compared our findings to published AAQ-24 studies involving midlife and later-life adults. Three-quarters of our respondents were women. There are no published sex-specific comparisons with which to compare our findings. Further, there are no published studies of AAQ-SFs reporting findings about non-heterosexual adults. There are published findings about associations between the AAQ-24 and other markers of interest during this COVID-19 recovery phase, including physical and mental health functioning [98,99,100]. We had no such measures to help us explore the merits of the AAQ-SF, and so these adjunct measures warrant future attention. The present study findings about adults in their 50s support the four-item PC and PL structure of the AAQ-SF and qualitative inquiry to better understand meanings and relevance within the PG subscale [63]. Further evidence is needed to confirm its validity with two independently studied international samples [1,101,102], multiple goodness-of-fit indices, prevailing theories, and the empirical literature at hand [1,40,46,49]. We hope that our findings will serve as an impetus for researchers in Canada and other countries to further assess and compare the psychometric properties of the AAQ-SF with other measures used with middle-aged adults. Patterns of response within and associations between the PG domain and other measures of mental health and well-being would provide added insight. Tools that measure age-related beliefs in this age group within and beyond the context of a pandemic are needed now and for future crises [91], if not everyday life in the years ahead.

## Figures and Tables

**Figure 1 ijerph-20-07035-f001:**
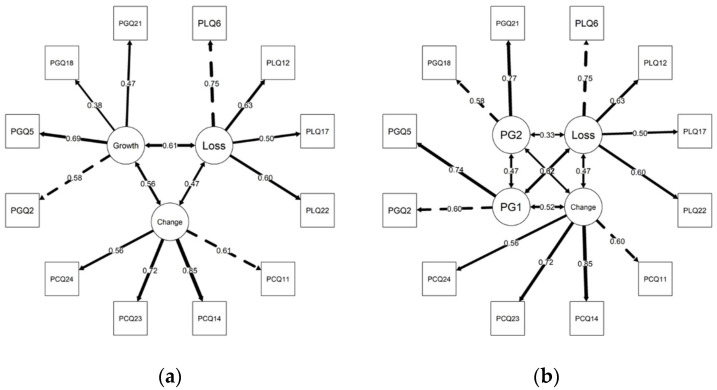
(**a**) Attitudes to Aging Questionnaire Short Form model with growth as a single subscale (*n* = 517). Growth = psychological growth; change = physical change; loss = psychological loss; (**b**) Attitudes to Aging Questionnaire Short Form alternative model with growth as a two-subscale structure (*n* = 517). Change = physical change; loss = psychological loss; PG1 = self; PG2 = generativity.

**Figure 2 ijerph-20-07035-f002:**
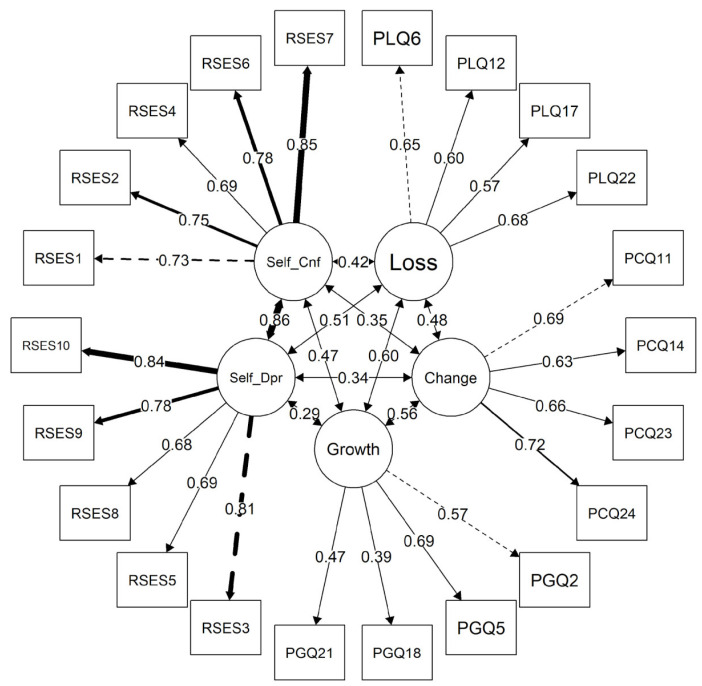
Convergent validity model (*n* = 517). Growth = psychological growth, change = physical change; loss: psychological loss; Self_Cnf = self-confidence; Self_Depr = self-deprecation.

**Table 1 ijerph-20-07035-t001:** Descriptive statistics for respondents’ AAQ-SF scores (*n* = 517).

AAQ-SF Subscale	M (SD)	Skew	Kurtosis	r	ω [95% CI]
Physical Change (PC)	13.45 (3.46)	−0.235	−0.262		0.78 (0.75–0.81)
I don’t feel old	3.79 (1.05)	−0.695	−0.075	0.692	
I have more energy now than I expected for my age	3.03 (1.11)	0.117	−0.934	0.831	
My health is better than expected for my age	3.19 (1.12)	−0.230	−0.679	0.822	
I meep myself as fit and active as possible by exercising	3.44 (1.18)	−0.466	−0.654	0.745	
Psychosocial Loss (PL)	14.42 (3.07)	−0.349	0.064		0.71 (0.68–0.76)
Old age is a depressing time of life	3.48 (0.99)	−0.16	−0.65	0.750	
I see old age mainly as a time of loss	3.70 (1.01)	−0.62	−0.01	0.717	
As I get older, I find it more difficult to make new friends	3.57 (1.09)	−0.63	−0.26	0.725	
I feel excluded from things because of my age	3.67 (1.06)	−0.67	−0.09	0.756	
Psychological Growth (PG)	15.23 (2.49)	−0.514	1.40		0.61 (0.57–0.67)
It is a privilege to grow old	3.90 (0.935)	−0.945	0.664	0.727	
There are many pleasant things about growing older	3.74 (0.857)	−0.920	1.068	0.667	
It is very important to pass on the benefits of my experiences to younger people	3.58 (0.919)	−0.362	0.013	0.664	
I want to give a good example to younger people	4.00 (0.831)	−0.944	1.334	0.685	

Note: AAQ-SF = Attitudes to Aging Questionnaire Short Form; M = mean; SD: standard deviation; r = Pearson’s moment correlation coefficients for items and respective parent subscales; ω = McDonald’s omega; CI = confidence interval.

**Table 2 ijerph-20-07035-t002:** Goodness-of-fit indices for the confirmatory factor analyses of the Attitudes to Aging Questionnaire Short Form with psychological growth as a single subscale and as an alternative two-subscale domain (*n* = 517).

Model	*X* ^2^	df	CFI	TLI	RMSEA [90% CI]
One-Subscale Growth	159.89	51	0.980	0.974	0.039 [0.025–0.052]
Two-Subscale Growth	116.14	48	0.927	0.900	0.05 [0.04–0.065]

Note: all chi-square tests are significant at *p* < 0.001; df = degrees of freedom; CFI = comparative fit index; TLI = Tucker–Lewis index; RMSEA = root mean square error of approximation; CI = confidence interval; growth = psychological growth.

**Table 3 ijerph-20-07035-t003:** Descriptive statistics for AAQ-SF scores across respondent subgroups (*n* = 517).

AAQ-SF Subscale	M (SD)	M (SD)	t
	Men (*n* = 108)	Women (*n* = 409)	
Physical Change	13.68 (3.63)	13.39 (3.41)	−0.774
Psychosocial Loss	14.86 (2.94)	14.31 (3.10)	−1.69
Psychological Growth	15.39 (2.91)	15.19 (2.37)	−0.736
	Post-secondary (*n* = 419)	No Post-secondary (*n* = 98)	
Physical Change	13.60 (3.42)	12.78 (3.58)	−2.11
Psychosocial Loss	14.35 (3.07)	14.68 (3.08)	0.951
Psychological Growth	15.29 (2.37)	14.98 (2.93)	−1.12
	Married/Partnered (*n* = 345)	Alone (*n* = 172)	
Physical Change	13.61 (3.45)	13.12 (3.45)	−6.55 ***
Psychosocial Loss	14.65 (2.88)	13.95 (3.40)	−3.49 **
Psychological Growth	15.35 (2.56)	14.99 (2.32)	1.18
	Has comorbidities (*n* = 243)	No comorbidities (*n* = 268)	
Physical Change	12.55 (3.34)	14.43 (3.13)	−6.55 ***
Psychosocial Loss	13.97 (2.98)	14.89 (3.00)	−3.49 **
Psychological Growth	15.15 (2.50)	15.40 (2.33)	−1.18
	Fair/Poor (*n* = 51)	Good to Excellent (*n* = 466)	
Physical Change	8.90 (2.71)	13.94 (3.16)	−10.97 ***
Psychosocial Loss	12.41 (3.59)	14.64 (2.93)	−5.02 ***
Psychological Growth	13.76 (2.78)	15.39 (2.40)	−4.52 ***

Note: *** t-statistics are significant at *p* < 0.001. **: t-statistics are significant at *p* < 0.01. AAQ-SF = Attitudes to Aging Questionnaire Short Form; M = mean; SD: standard deviation.

**Table 4 ijerph-20-07035-t004:** Goodness-of-fit indices for the confirmatory factor analysis (*n* = 517).

Model	*X* ^2^	Df	CFI	TLI	RMSEA [90% CI]
One-Subscale Growth	454.12	205	0.866	0.849	0.049 [0.043–0.055]

Note: chi-square test is significant at *p* < 0.001; df = degrees of freedom; CFI = comparative fit index; TLI = Tucker–Lewis index; RMSEA = root mean square error of approximation; CI = confidence interval; growth = psychological growth.

## Data Availability

The datasets generated during and/or analyzed during the current study are available from the corresponding author on reasonable request.

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
