# Peer review of "Suitability of the Attitudes to Aging Questionnaire Short Form for Use among Adults in Their 50s: A Cross-Sectional e-Survey Study"

_ijerph, 2023, doi:10.3390/ijerph20227035_

Round 1

Reviewer 1 Report

Comments and Suggestions for Authors

The authors’ aim was to determine the suitability for the use of the 12-item Attitudes to Aging Questionnaire Short-Form (AAQ-SF) among Canadian people in their 50’s. They also ask to the participants to respond to the Rosenberg Self-Esteem Scale and a 16 short sociodemographic profile. An e-survey was carried out on a sample of 517 middle-aged adults. After a categorical confirmatory factor analysis, convergent validity, findings suggest the suitability of the AAQ-SF for people on their 50`s for the Physical Change & Physical Loss subscales. Their findings also suggest a two-factor structure for the Psychological Growth subscale.

Comments

Title. The study`s design is included in the abstract, as a suggestion authors may consider adding it in the title. Consider using as e-survey.

Abstract. It provides an informative and balanced summary of what was done and was found.

Introduction. The report successfully explains the aim of the study and the background for the investigation. Rationale has been well reported.

Methods. The target population is well described. A convenience sample was performed, may include if it was an open or closed sample. Timeframe is included.

Although they commented about the informed consent inclusion, they failed to mention about the protection of personal information mechanism (collect and store).

They don´t include information about the previous testing of the electronic questionnaire. On the other hand, authors do include the contact mode for the recruitment process and the description of the access to the questionnaire. Ideally the survey announcement/mailed should be published as an appendix.

It is a suggestion to include that it was a voluntary survey and did not offer monetary or non-monetary incentives. Authors included the questionnaire participation rate but did not give information on the completion rate, or how do they prevent multiple entries from the same individual.

Measures were well described, number of items, sub-scales, their names, and items as well as the scoring format and the number of response alternatives. They also point about the items that are reverse scored.

Statistical approach to assess the basic statistical analysis, the reliability of the AAQ-SF, the discriminative validity, the confirmatory factor analysis, chi-square model and convergent validity are all well described. The authors clearly reported the software and version used for their calculations.

Results. Authors include sufficient information to get a picture of the participants. All the analysis are well reported.

Discussion. A successful and cautious overall interpretation of the results, summarizing key results and discussing them with reference to the study aims and the results from other studies, limitations, and relevant evidence to suggest further improvements or paths of study.

They conclude in reference to the main objective of the study and exposed funding information and no conflict of interests.

No further comments

Author Response

The authors’ aim was to determine the suitability for the use of the 12-item Attitudes to Aging Questionnaire Short-Form (AAQ-SF) among Canadian people in their 50’s. They also ask to the participants to respond to the Rosenberg Self-Esteem Scale and a 16 short sociodemographic profile. An e-survey was carried out on a sample of 517 middle-aged adults. After a categorical confirmatory factor analysis, convergent validity, findings suggest the suitability of the AAQ-SF for people on their 50`s for the Physical Change & Physical Loss subscales. Their findings also suggest a two-factor structure for the Psychological Growth subscale.

Comments

1. Title. The study`s design is included in the abstract, as a suggestion authors may consider adding it in the title. Consider using as e-survey.

We thank you for this suggestion. I have added 'cross-sectional e-survey study' to the title. Please note that we have also added this terminology into Section 2.1 Study Design and Participants: “In this cross-sectional e-survey study, data were collected in July and August 2019 by Zoomer Media, an affiliate of the Canadian Association of Retired Persons or CARP.”

Abstract. It provides an informative and balanced summary of what was done and was found.

Introduction. The report successfully explains the aim of the study and the background for the investigation. Rationale has been well reported.

Methods. The target population is well described. A convenience sample was performed, may include if it was an open or closed sample. Timeframe is included.

2. Although they commented about the informed consent inclusion, they failed to mention about the protection of personal information mechanism (collect and store).

As we now indicate in Section 2.3 Data Collection Procedure [paragraph 1], “To ensure full anonymity, the names, IP addresses, and therefore the number of persons who initially received the digital advertisement were not shared with any study investigator” and “All study data were fully encrypted and stored on a password protected computer in the primary investigator’s department.”

3. They don't include information about the previous testing of the electronic questionnaire. On the other hand, authors do include the contact mode for the recruitment process and the description of the access to the questionnaire. Ideally the survey announcement/mailed should be published as an appendix.

We thank you for bringing this to our attention. As noted in section 2.2, we draw attention to “the e-blast (Appendix A).” Appendix A contains our study advertisement sent out in the e-blast.

4. It is a suggestion to include that it was a voluntary survey and did not offer monetary or non-monetary incentives.

We now indicate in Section 2.3 Data Collection Procedure [paragraph 1]: “The survey involved a voluntary e-survey. No incentive, albeit monetary or non-monetary, was offered. Recipients who clicked on the survey hyperlink were first taken to a study information letter and an informed consent form which asked: “I agree to participate in the research study described: yes/no.”

5. Authors included the questionnaire participation rate but did not give information on the completion rate, or how do they prevent multiple entries from the same individual.

We have amended the statement in Section 2.3 Data Collection Procedure [paragraph 1] to read as follows: “Among the 779 persons who clicked the survey hyperlink, as tallied by Survey Monkey©, 517 (66.37%) were “yes” responders who also went on to complete our study questionnaires.”   

In Section 2.3 Data Collection Procedure [paragraph 1], we now further remark “Survey Monkey© advanced survey logic prevented the same responder from taking our survey more than once.”

Measures were well described, number of items, sub-scales, their names, and items as well as the scoring format and the number of response alternatives. They also point about the items that are reverse scored.

Statistical approach to assess the basic statistical analysis, the reliability of the AAQ-SF, the discriminative validity, the confirmatory factor analysis, chi-square model and convergent validity are all well described. The authors clearly reported the software and version used for their calculations.

Results. Authors include sufficient information to get a picture of the participants. All the analysis are well reported.

Discussion. A successful and cautious overall interpretation of the results, summarizing key results and discussing them with reference to the study aims and the results from other studies, limitations, and relevant evidence to suggest further improvements or paths of study.

They conclude in reference to the main objective of the study and exposed funding information and no conflict of interests.

6. No further comments

We thank reviewer #1 for their thoughtful suggestions and feedback.

Reviewer 2 Report

Comments and Suggestions for Authors

Thank you for reporting the findings of this very important study. I appreciate that the paper is generally well-written, but I have listed some minor points below. We would appreciate your consideration.

1. In title, the design of the study is presented using the study’s design with a commonly used term.

2. Throughout the manuscript, it is difficult to understand what the term "suitability" refers to. Does it refer to reliability and validity? If it is to be used in the title, it should be specifically defined.

3. In 2.2 procedure, Since the attributes of the population for this sample are not clear, please explain with reference to specific data.

4. Could you give the eligibility criteria of participants.

5. Please detail how the sample size was determined. Also describe why the random sampling was done with three provinces.

6. For each of the five rating of the AAQ-SF, please detail how each rating was described.

Author Response

Thank you for reporting the findings of this very important study. I appreciate that the paper is generally well-written, but I have listed some minor points below. We would appreciate your consideration.

1. In title, the design of the study is presented using the study’s design with a commonly used term. 

In keeping with reviewer #1’s request, we have added the word ‘e-survey’ into the title.

2. Throughout the manuscript, it is difficult to understand what the term "suitability" refers to. Does it refer to reliability and validity? If it is to be used in the title, it should be specifically defined.

We thank you for bringing this to our attention. We further clarify in the Abstract: “This cross-sectional e-survey study examines the suitability (reliability and validity) of the 12-item Attitudes to Aging Questionnaire Short-Form (AAQ-SF) for use among adults in their 50s.”

In Section 4 Discussion [paragraph 1], we also remind the reader of the following: “This Canadian study attempted to determine the suitability (reliability and validity) of the AAQ-SF for use among people in their 50s.”

3. In 2.2 procedure, Since the attributes of the population for this sample are not clear, please explain with reference to specific data.

In response to this important point, we have restructured Section 2 into 2.1 through to 2.4, with this also improving flow.

In Section 2.1 Study Design and Participants, we therefore now state: “Canadians can become CARP members as early as age 45 (average age = 61 years of age) and can become Zoomer subscribers as early as age 35 (average age = 57 years of age) [30,31]. Other characteristics among CARP members taking part in national polls the organization conducted are: 55% males; 74% married/common law; average household income: $130,000; and 77% lived in households with two to five or more persons [30]. Characteristics of Zoomer subscribers are: 44% males; average age of 57; and average household income: $78,556 [31]. About 61% of CARP members have a post-secondary degree [30], as do about 54% of Zoomer subscribers [32]. Marital status and numbers of household members are not available for Zoomers.”

Section 2.2 contains ‘Eligibility Criteria’. Section 2.3 speaks to our ‘Data Collection Procedure’. Section 2.4 is about our ‘Statistical Analyses’.  

4. Could you give the eligibility criteria of participants.

We now point out under Section 2.2 Eligibility Criteria that “Eligibility criteria were: (1) being 50 to 59 years of age; (2) having provided CARP/Zoomer magazine an email address to receive communications like surveys; (5) freely consenting to participate in this survey.”

5. Please detail how the sample size was determined.

Acknowledging the ongoing discussions about suitable sample sizes in SEM modeling, the literature hasn't reached a consensus on exact requirements [43,44,49]. Some experts suggest that a "substantial" sample, typically over 300, is needed for SEM. Morrison et al. [49] drew attention to Soper’s (2006-2023) SEM sample size calculator (https://www.danielsoper.com/statcalc/calculator.aspx?id=89).

Accordingly, in Section 2.4 Statistical Analysis [paragraph 6], we now state “A minimum sample size was calculated using Soper’s online SEM sample size calculator. In keeping with our CFA our loading cut-off criterion [37] for the AAQ-SF (3 latent variables and 12 observed variables, a desired power of 80% and a statistical significance criterion of α = .05), our required sample size was n = 100 [51]. For the convergent validity analysis, given our cut-off criterion [50] for observed associations between the AAQ-SF and RSES (2 latent variables and 10 observed variables), it was n = 150 [51].”

In Section 3 Results [paragraph 4], we also add “Our sample size was sufficient for testing a four latent variable structure [51].”

6. Also describe why the random sampling was done with three provinces.

We thank you for raising this question. The primary investigator has clarified this with our data collectors. When our study commenced, CARP members resided mainly in Ontario, British Columbia, and Alberta. Zoomer members largely resided in Ontario, and others in BC and Alberta. Collecting data in these three provinces was purposive, not random, to maximize our responder pool. We sincerely apologize for this oversight.

Accordingly, in Section 2.3 Data Collection Procedure, we now state: “An e-blast (Appendix A) about the study aim, team, and a survey hyperlink was sent to CARP members and Zoomer subscribers in the three provinces in which most lived: Ontario, British Columbia, and Alberta [30,31]. We have also added to Section 5 Limitations and Conclusions: “Our study was conducted only 3 of 10 10 provinces in a single country among a single 10-year age group.”

7. For each of the five rating of the AAQ-SF, please detail how each rating was described.

We thank you for this request. In Section 2.3 under ‘Measures’, ‘Attitudes to Aging Questionnaire - Short-Form (AAQ-SF, [1])’ we provide greater clarity: “Self-report ratings for items 1 to 3 range from 1 to 5, where 1 reflects strongly disagree, 2 reflects disagree, 3 reflects uncertain, 4 reflects agree, and 5 reflects strongly agree. Response categories for items 4 to 12 are: 1 = not at all true; 2 = slightly true; 3 = moderately true; 4 = very true; and 5 = extremely true. The four negatively worded PL items (item 3: ‘Old age is a depressing time of life’; item 5: ‘I see old age mainly as a time of loss’; item 7: ‘As I get older, I find it difficult to make new friends’; item 10: ‘I feel excluded from things because of my age’) are reverse coded so that higher scores reflect more positive attitudes.”

The insightful suggestions from Reviewer #2 very much strengthens our paper.

Reviewer 3 Report

Comments and Suggestions for Authors

The manuscript focuses on the psychometric properties of the 12-item Attitudes to Aging Questionnaire Short-Form (AAQ-SF) for use among adults. Researchers and health professionals may be interested in the results that were found and reported.

I find the introduction to be well written and grounded, which is a good summary of attitudes to ageing and the AAQ-SF.

My major criticism is the methodology for conducting the factor analysis of instruments:

The focus of the study is to examine the factor structure of the AAQ-SF, however, the results of the confirmatory factor analysis do not seem conclusive. For the model with a good fit (1a), the authors test another model (1b), where the factor weights are better but the fit decreases, which does not confirm the validity of the structural model. CFI and TLI indicators below 0.95 are not really considered good or acceptable. It would be very important to compare these results with other international results, e.g., Yeh et al. 2022. It would be crucial to take note that there is presumably a strong methodological effect here, which is due to the positively and negatively formulated items, which it is not convenient to isolate as factors. The same is the case with Rosenberg's Self-Esteem questionnaire, where researchers usually do not talk about two factors but about a single dimension and a method factor. Awareness of this feature would be extremely important, as it could guide further research. In my opinion, it would be worthwhile to use exploratory factor analysis to look at the factor structure and determine how many dimensions the measure has (e.g., parallel analysis).

The analysis is not fully consistent with the descriptions. The factor structure analysis should be presented first, followed by group comparisons, and so on. In my opinion, the illustration of skewness and kurtosis for each item (not only for scales) could also be important because it helps to judge the method of factor analysis.
In the case of RSES, the fit indicators are weak and indicate that the two-dimensional model is not adequate, so the Convergent validity model in Figure 2 is not accurate.
So, overall, I feel the manuscript is an important initiative, but the analysis and interpretation of the results are imprecise. In my opinion, it would be worthwhile to focus only on the factor structure of the AAQ-SF, with exploratory and then confirmatory factor analysis, while also considering international results more carefully (e.g., positively and negatively worded items). In addition to this, it would be important to provide a more in-depth interpretation of the results of factor analysis in the introductory section.

Yeh, C. J., Yang, C. Y., Lee, S. H., Wang, C. Y., & Lee, M. C. (2022). Psychometric Properties of a Taiwanese Attitude toward Aging Questionnaire-Short Form Used in the Taiwan Longitudinal Study on Aging. International Journal of Gerontology16(3).

Author Response

The manuscript focuses on the psychometric properties of the 12-item Attitudes to Aging Questionnaire Short-Form (AAQ-SF) for use among adults. Researchers and health professionals may be interested in the results that were found and reported. I find the introduction to be well written and grounded, which is a good summary of attitudes to ageing and the AAQ-SF.

My major criticism is the methodology for conducting the factor analysis of instruments:

1. The focus of the study is to examine the factor structure of the AAQ-SF, however, the results of the confirmatory factor analysis do not seem conclusive. For the model with a good fit (1a), the authors test another model (1b), where the factor weights are better but the fit decreases, which does not confirm the validity of the structural model. CFI and TLI indicators below 0.95 are not really considered good or acceptable. It would be very important to compare these results with other international results, e.g., Yeh et al. 2022. 

We thank Reviewer #3 for their emphasis on fit indices. We now point out in Section 2.4 Statistical Analyses [paragraph 3]: “Adjunct goodness-of-fit assessments were: (1) Comparative Fit Index (CFI) and (2) Tucker-Lewis Index (TLI), with values over .90 recommended [37,42] and with .95 as ideal [43].”

We agree that we cannot fully confirm the validity of the structural model. As Laidlaw and colleagues [1] point out “data are not perfect” (p. 118) thus along with multiple indices of fit multi-country samples are recommended. Marsh and colleagues offer another perspective in that “if theory predicts that a path coefficient should be positive, whereas the observed result shows it is negative, high levels of GOF are not sufficient to argue for the validity of predictions based on a model (p. 324) [44].

Therefore in Section 4 Discussion [paragraph 5] we add the following “Overall, this alternative structure exhibited reasonable fit with our data, although not as robust as a model with CFI and TLI values over 0.90 [1,37,42,43]. The RMSEA suggests a close fit [37,42,43]. Goodness-of-fit indices are but one yardstick with which to judge the validity of a model [42]. There are theoretical and empirical reasons for ‘reasonable fit’.”

In Section 5 Limitations and Conclusions [paragraph 1], we also now tell readers that “The present study findings about adults in their 50s support the 4-item PC and PL structure of the AAQ-SF and a two-factor structure for the PG subscale. Further evidence is needed to fully confirm its validity, ideally with an international sample [1] and multiple goodness-of-fit indices, prevailing theories, and empirical literature at hand [1, 38,44,49].”

We incorporate Yeh et al.’s insightful findings in Section 4 Discussion [paragraph 9] and with respect to model fit in convergent validity assessments.

2. It would be crucial to take note that there is presumably a strong methodological effect here, which is due to the positively and negatively formulated items, which it is not convenient to isolate as factors. The same is the case with Rosenberg's Self-Esteem questionnaire, where researchers usually do not talk about two factors but about a single dimension and a method factor. Awareness of this feature would be extremely important, as it could guide further research.

We thank Reviewer #3 for bringing these important points to our attention. WHOQOL instruments are rigorously scrutinized through multi-country focus groups, pilot studies with debriefing, and field trials to reduce problematic item wording and patterns of response. Nonetheless negative item method bias [63-64] is a possibility. Post-COVID, we continue to learn about the behaviors of links between own aging perceptions and mental health and well-being across countries [85]. We also want policy makers and practitioners to duly consider the merits of the AAQ-SF, given the stresses and strains that midlife adults have endured during the pandemic. In keeping with IJEPHR’s spirit of open access, looking at self-confidence and self-deprecation should appeal to a variety of readers.

We therefore now add this to Section 4.0. Discussion [paragraph 9]: “We are also cognizant that people’s preferences for positive emotional states can increase with age [54-56]. For example, older people with higher life satisfaction scores have been shown to be more likely to endorse self-confidence items on the RSES [63-64]. In a recent study of midlife adults, those tending to be more confident about self-managing chronic illness had higher scores on positively worded items on the Taiwanese AAQ-SF [65]. Among the midlife adults in our study, PG scores were highest overall and most strongly correlated with self-confidence. The debate over whether the RSES best fits one construct with positively worded items or two (positively and negatively worded) remains unresolved [66-67]. Choices regarding dimensionality are based on other fundamental considerations [38,44,49]. Some advise that if a theory or previous empirical findings suggest that positively and negatively oriented items measure distinct constructs, treat them accordingly [66]. Our two-subscale approach should also have practical appeal. An instrument’s ability to capture multiple aspects of people’s lives has long been considered most helpful for policy makers and practitioners, and the populations with whom they work [68]. How people see their own process of aging can impact mental health for them in their 50s and well into later life [16]. Knowing whether opportunities for PG are largely tied to self-confidence, not self-deprecation, is therefore important, particularly in post-pandemic times.”

In Section 5 Limitations and Conclusions, we also now state “Patterns of response within and associations between the PG domain and other measures of mental health and well-being would provide added insight.”

3. In my opinion, it would be worthwhile to use exploratory factor analysis to look at the factor structure and determine how many dimensions the measure has (e.g., parallel analysis). 

Thank you for your feedback. Confirmatory factorial models can be directly applied when there is a robust theoretical foundation based on previous studies (DiStefano & Hess, 2005; Schmitt, 2011). Moreover, as stated by Orçan (2018), an Exploratory Factor Analysis is necessary when seeking evidence of validity for the internal structure of new measures or those undergoing cultural adaptation.

For the parent AAQ-24, goodness-of-fit was superior for a three-factor structure versus a single-dimension or three-factor structure loading onto a fourth higher-order factor and across a 20-country sample [2]. A good number of other AAQ-24 studies in different countries, including Canada in later life [9] and at midlife [33,34], supporting a three-factor structure had been cited in our August 29th manuscript.

In Section 1 Introduction [paragraph 2], as we focus on the AAQ-SF, we now state “The superior goodness-of-fit of the AAQ-SF’s three-factor versus single factor structure and no less superior fit than a three-factor structure loading onto a fourth higher-order factor was observed among an at-random split-half 20-country sample of some 2400 responders, and further confirmed among a third independent studied sample (n = 792) [1].”

In turn, we offer DiStefano & Hess; Schmitt; and Orcan as food-for-thought.

DiStefano, C.; Hess, B. (2005). Using Confirmatory Factor Analysis for Construct Validation: An Empirical Review. J. Psychoeduc.l Assessment 2005, 23, 225–241. https://doi.org/10.1177/073428290502300303

Schmitt, T.A. Current Methodological Considerations in Exploratory and Confirmatory Factor Analysis. J. Psychoeduc. Assessment 201129, 304–321. https://doi.org/10.1177/0734282911406653

Orçan, F. Exploratory and confirmatory factor analysis: which one to use first? J. Measurement and Eval. Educ. and Psychol. 2018, 9(4), 414-421. https://doi.org/10.21031/epod.394323

4. The analysis is not fully consistent with the descriptions. In Section 3 Results, the factor structure analysis should be presented first, followed by group comparisons, and so on.

We thank Reviewer #3 for bringing the matter of order of presentation to our attention. In Section 2.3 Statistical Analyses we now first speak to factor structure, followed by group comparisons and convergent validity. Relatedly, in Section 3 Results we present our factor structure first (Table 2, Figures 1(a) and 1(b)), followed by group comparisons (Table 3) and, thereafter, our convergent validity analysis (Table 4). These adjustments also better fit with the flow of our discussion.

5. In my opinion, the illustration of skewness and kurtosis for each item (not only for scales) could also be important because it helps to judge the method of factor analysis.

Thank you for your observation.  We have now added the skewness and kurtosis for each item in Table 1. In Section 2.4 Statistical Analysis, we alert the reader to the following: “A categorical (ordinal) confirmatory factor analysis (CFA) was conducted of the proposed 3-subscale 12-item AAQ-SF structure [1] using a Weighted Least Square Mean and Variance Adjusted or WLSMV estimation method based on polychoric correlations due to the categorical nature of item responses [34,35].”

6. In the case of RSES, the fit indicators are weak and indicate that the two-dimensional model is not adequate, so the Convergent validity model in Figure 2 is not accurate.

Thank you for raising this question and concern. A recent study [66] investigating the effect of using a negatively oriented Rosenberg Self Esteem Scale and two other scales, in both original and reversed conditions, supports a multidimensional understanding when both positively and negatively oriented items are present (i.e., the better fit of the two-factor model compared to the one-factor model for all three tools). Thus, even though we can view the data as unidimensional, our model designs are consistent with earlier indications of multidimensionality.

7. So, overall, I feel the manuscript is an important initiative, but the analysis and interpretation of the results are imprecise.

We trust that the changes we have made in response to your queries address this concern.

8. In my opinion, it would be worthwhile to focus only on the factor structure of the AAQ-SF, with exploratory and then confirmatory factor analysis, while also considering international results more carefully (e.g., positively and negatively worded items).

Thank you for your feedback. We address this under Point #2 and Point #3. I will share your perspectives with Dr. Kenneth Laidlaw.

9. In addition to this, it would be important to provide a more in-depth interpretation of the results of factor analysis in the introductory section.

We concur. Please see our response under point #3.

Round 2

Reviewer 3 Report

Comments and Suggestions for Authors

Thank you for your careful corrections and I am happy to accept the manuscript. Perhaps the issue of the CFI and TLI fit indicators should be clarified. Also, I forgot to point out earlier that it is unfortunate to use abbreviations (AAQ-SF) in the title. 

The manuscript focuses on the psychometric properties of the 12-item Attitudes to Aging Questionnaire Short-Form (AAQ-SF) for use among adults. Researchers and health professionals may be interested in the results that were found and reported. I find the introduction to be well written and grounded, which is a good summary of attitudes to ageing and the AAQ-SF.

My major criticism is the methodology for conducting the factor analysis of instruments:

1. The focus of the study is to examine the factor structure of the AAQ-SF, however, the results of the confirmatory factor analysis do not seem conclusive. For the model with a good fit (1a), the authors test another model (1b), where the factor weights are better but the fit decreases, which does not confirm the validity of the structural model. CFI and TLI indicators below 0.95 are not really considered good or acceptable. It would be very important to compare these results with other international results, e.g., Yeh et al. 2022.

We thank Reviewer #3 for their emphasis on fit indices. We now point out in Section 2.4 Statistical Analyses [paragraph 3]: “Adjunct goodness-of-fit assessments were: (1) Comparative Fit Index (CFI) and (2) Tucker-Lewis Index (TLI), with values over .90 recommended [37,42] and with .95 as ideal [43].”

References 37 and 42 are misleading. Hair et al (2019) state in their study: 

„Assessing Fit

Fit was discussed in detail in Chapter 9. Recall that reality is represented by a covariance matrix of measured items (S), and the theory is represented by the proposed measurement model structure. Equations are implied by the model, as discussed earlier in this chapter and in Chapter 9. The equations enable us to estimate reality by computing an estimated covariance matrix based on our theory (Σk). Fit compares the two covariance matrices. Guidelines for good fit provided in Chapter 9 apply. Here the researcher attempts to examine all aspects of construct validity through various empirical measures. The result is that CFA enables us to test the validity of a theoretical measurement model. CFA is quite different from EFA, which explores data to identify potential constructs. Many researchers conduct EFA on one or more separate samples before reaching the point of trying to confirm a model. EFA is an appropriate tool for identifying factors among multiple variables. As such, EFA results can be useful in helping to develop theory that will lead to a proposed measurement model. After a researcher builds confidence in a measurement model, CFA enters the picture.”

And in chapter 9 you'll find this:

„Cut-off Values for Fit Indices: The Magic .90, or Is that .95?

Although we know we need to complement the x2 with additional fit indices, one question still remains no matter what index is chosen: What is the appropriate cut-off value for that index? For most of the incremental fit statistics, accepting models producing values of .90 became standard practice in the early 1990s. However, some scholars concluded that .90 was too low and could lead to false conclusions, and by the end of the decade .95 had become the recommended standard for indices such as the TLI and CFI [25]. In general, .95 somehow became the magic number indicating good-fitting models, even though no empirical evidence supported such a development. Research has challenged the use of a single cut-off value for GOF indices, finding instead that a series of additional factors can affect the index values associated with acceptable fit. First, research using simulated data (for which the actual fit is known) provides counterarguments to these cut-off values and does not support .90 as a generally acceptable rule of thumb [25]. It demonstrates that at times even an incremental goodness-of-fit index above .90 would still be associated with a severely misspecified model. This suggests that cut-off values should be set higher than .90. Second, research continues to support the notion that model complexity unduly affects GOF indices, even with something as simple as just more indicators per construct [31]. Finally, the underlying distribution of data can influence fit indices [16]. In particular, as data become less appropriate for the particular estimation technique selected, the ability of fit indices to accurately reflect misspecification can vary. This issue seems to affect incremental fit indices more than absolute fit indices. What has become clear is that no single “magic” value always distinguishes good models from bad models. GOF must be interpreted in light of the characteristics of the research. It is interesting to compare these issues in SEM to the general lack of concern for establishing a magic R2  number in multiple regression. If a magic minimum R2 value of .5 had ever been imposed, it would be just an arbitrary limit that would exclude potentially meaningful research. So we should be cautious in adopting one size fits all standards. It is simply not practical to apply a single set of cut-off rules that apply for all SEM models of any type.”

For further reference, see the other 2 articles: Goretzko et al., 2023; McNeish & Wolf, 2023.

And in the book cited in reference 42, there is no clear criterion for CFI and TLI, but Byrne decides which is the better model based on the fit indices of competing models. In the case of AAQ-SF the opposite was the case, a model with weaker fit indices was adopted. Of course, I do not think that the validity of the psychometric model is more important than that of the psychological theory, but the acceptability of the fit indices is misleading and should not be used as a justification. It is acceptable to me that the fit indicator is lower, but it is more psychologically explainable (as you described below). I think that the study by Wolf (2023) can be a good help here.

References

Goretzko, D., Siemund, K., & Sterner, P. (2023). Evaluating Model Fit of Measurement Models in Confirmatory Factor Analysis. Educational and Psychological Measurement, 00131644231163813.

Hair, J.F.; Black, W.C.; Babin, B.; Anderson, R.E. SEM: Confirmatory factor analysis. In Multivariate Data Analysis, 8th ed. Cengage Learning: Andover, Hampshire, United Kingdom, 2019, pp. 658-698; ISBN 978-1-4737-5654-0.

McNeish, D., & Wolf, M. G. (2023). Dynamic fit index cutoffs for confirmatory factor analysis models. Psychological Methods, 28(1), 61.

Wolf, M. G. (2023). The problem with over-relying on quantitative evidence of validity. https://psyarxiv.com/v4nb2/download?format=pdf

We agree that we cannot fully confirm the validity of the structural model. As Laidlaw and colleagues [1] point out “data are not perfect” (p. 118) thus along with multiple indices of fit multi-country samples are recommended. Marsh and colleagues offer another perspective in that “if theory predicts that a path coefficient should be positive, whereas the observed result shows it is negative, high levels of GOF are not sufficient to argue for the validity of predictions based on a model (p. 324) [44].

Therefore in Section 4 Discussion [paragraph 5] we add the following “Overall, this alternative structure exhibited reasonable fit with our data, although not as robust as a model with CFI and TLI values over 0.90 [1,37,42,43]. The RMSEA suggests a close fit [37,42,43]. Goodness-of-fit indices are but one yardstick with which to judge the validity of a model [42]. There are theoretical and empirical reasons for ‘reasonable fit’.”

In Section 5 Limitations and Conclusions [paragraph 1], we also now tell readers that “The present study findings about adults in their 50s support the 4-item PC and PL structure of the AAQ-SF and a two-factor structure for the PG subscale. Further evidence is needed to fully confirm its validity, ideally with an international sample [1] and multiple goodness-of-fit indices, prevailing theories, and empirical literature at hand [1, 38,44,49].”

We incorporate Yeh et al.’s insightful findings in Section 4 Discussion [paragraph 9] and with respect to model fit in convergent validity assessments.

2. It would be crucial to take note that there is presumably a strong methodological effect here, which is due to the positively and negatively formulated items, which it is not convenient to isolate as factors. The same is the case with Rosenberg's Self-Esteem questionnaire, where researchers usually do not talk about two factors but about a single dimension and a method factor. Awareness of this feature would be extremely important, as it could guide further research.

We thank Reviewer #3 for bringing these important points to our attention. WHOQOL instruments are rigorously scrutinized through multi-country focus groups, pilot studies with debriefing, and field trials to reduce problematic item wording and patterns of response. Nonetheless negative item method bias [63-64] is a possibility. Post-COVID, we continue to learn about the behaviors of links between own aging perceptions and mental health and well-being across countries [85]. We also want policy makers and practitioners to duly consider the merits of the AAQ-SF, given the stresses and strains that midlife adults have endured during the pandemic. In keeping with IJEPHR’s spirit of open access, looking at self-confidence and self-deprecation should appeal to a variety of readers.

We therefore now add this to Section 4.0. Discussion [paragraph 9]: “We are also cognizant that people’s preferences for positive emotional states can increase with age [54-56]. For example, older people with higher life satisfaction scores have been shown to be more likely to endorse self-confidence items on the RSES [63-64]. In a recent study of midlife adults, those tending to be more confident about self-managing chronic illness had higher scores on positively worded items on the Taiwanese AAQ-SF [65]. Among the midlife adults in our study, PG scores were highest overall and most strongly correlated with self-confidence. The debate over whether the RSES best fits one construct with positively worded items or two (positively and negatively worded) remains unresolved [66-67]. Choices regarding dimensionality are based on other fundamental considerations [38,44,49]. Some advise that if a theory or previous empirical findings suggest that positively and negatively oriented items measure distinct constructs, treat them accordingly [66]. Our two-subscale approach should also have practical appeal. An instrument’s ability to capture multiple aspects of people’s lives has long been considered most helpful for policy makers and practitioners, and the populations with whom they work [68]. How people see their own process of aging can impact mental health for them in their 50s and well into later life [16]. Knowing whether opportunities for PG are largely tied to self-confidence, not self-deprecation, is therefore important, particularly in post-pandemic times.”

In Section 5 Limitations and Conclusions, we also now state “Patterns of response within and associations between the PG domain and other measures of mental health and well-being would provide added insight.”

3. In my opinion, it would be worthwhile to use exploratory factor analysis to look at the factor structure and determine how many dimensions the measure has (e.g., parallel analysis).

Thank you for your feedback. Confirmatory factorial models can be directly applied when there is a robust theoretical foundation based on previous studies (DiStefano & Hess, 2005; Schmitt, 2011). Moreover, as stated by Orçan (2018), an Exploratory Factor Analysis is necessary when seeking evidence of validity for the internal structure of new measures or those undergoing cultural adaptation.

For the parent AAQ-24, goodness-of-fit was superior for a three-factor structure versus a single-dimension or three-factor structure loading onto a fourth higher-order factor and across a 20-country sample [2]. A good number of other AAQ-24 studies in different countries, including Canada in later life [9] and at midlife [33,34], supporting a three-factor structure had been cited in our August 29th manuscript.

In Section 1 Introduction [paragraph 2], as we focus on the AAQ-SF, we now state “The superior goodness-of-fit of the AAQ-SF’s three-factor versus single factor structure and no less superior fit than a three-factor structure loading onto a fourth higher-order factor was observed among an at-random split-half 20-country sample of some 2400 responders, and further confirmed among a third independent studied sample (n = 792) [1].”

In turn, we offer DiStefano & Hess; Schmitt; and Orcan as food-for-thought.

DiStefano, C.; Hess, B. (2005). Using Confirmatory Factor Analysis for Construct Validation: An Empirical Review. J. Psychoeduc.l Assessment 2005, 23, 225–241. https://doi.org/10.1177/073428290502300303

Schmitt, T.A. Current Methodological Considerations in Exploratory and Confirmatory Factor Analysis. J. Psychoeduc. Assessment 201129, 304–321. https://doi.org/10.1177/0734282911406653

Orçan, F. Exploratory and confirmatory factor analysis: which one to use first? J. Measurement and Eval. Educ. and Psychol. 2018, 9(4), 414-421. https://doi.org/10.21031/epod.394323

The EFA could have been of help in arranging the items of the Psychological Growth scale (e.g. communality). The reliability of the 4-item scale is low, and the 2-2 item pairs are not really scales.

4. The analysis is not fully consistent with the descriptions. In Section 3 Results, the factor structure analysis should be presented first, followed by group comparisons, and so on.

We thank Reviewer #3 for bringing the matter of order of presentation to our attention. In Section 2.3 Statistical Analyses we now first speak to factor structure, followed by group comparisons and convergent validity. Relatedly, in Section 3 Results we present our factor structure first (Table 2, Figures 1(a) and 1(b)), followed by group comparisons (Table 3) and, thereafter, our convergent validity analysis (Table 4). These adjustments also better fit with the flow of our discussion.

5. In my opinion, the illustration of skewness and kurtosis for each item (not only for scales) could also be important because it helps to judge the method of factor analysis.

Thank you for your observation. We have now added the skewness and kurtosis for each item in Table 1. In Section 2.4 Statistical Analysis, we alert the reader to the following: “A categorical (ordinal) confirmatory factor analysis (CFA) was conducted of the proposed 3-subscale 12-item AAQ-SF structure [1] using a Weighted Least Square Mean and Variance Adjusted or WLSMV estimation method based on polychoric correlations due to the categorical nature of item responses [34,35].”

6. In the case of RSES, the fit indicators are weak and indicate that the two-dimensional model is not adequate, so the Convergent validity model in Figure 2 is not accurate.

Thank you for raising this question and concern. A recent study [66] investigating the effect of using a negatively oriented Rosenberg Self Esteem Scale and two other scales, in both original and reversed conditions, supports a multidimensional understanding when both positively and negatively oriented items are present (i.e., the better fit of the two-factor model compared to the one-factor model for all three tools). Thus, even though we can view the data as unidimensional, our model designs are consistent with earlier indications of multidimensionality.

7. So, overall, I feel the manuscript is an important initiative, but the analysis and interpretation of the results are imprecise.

We trust that the changes we have made in response to your queries address this concern.

8. In my opinion, it would be worthwhile to focus only on the factor structure of the AAQ-SF, with exploratory and then confirmatory factor analysis, while also considering international results more carefully (e.g., positively and negatively worded items).

Thank you for your feedback. We address this under Point #2 and Point #3. I will share your perspectives with Dr. Kenneth Laidlaw.

9. In addition to this, it would be important to provide a more in-depth interpretation of the results of factor analysis in the introductory section.

We concur. Please see our response under point #3.

Author Response

1. References 37 and 42 are misleading.

Hair et al (2019) state in their study: 

„Assessing Fit

Fit was discussed in detail in Chapter 9. Recall that reality is represented by a covariance matrix of measured items (S), and the theory is represented by the proposed measurement model structure. Equations are implied by the model, as discussed earlier in this chapter and in Chapter 9. The equations enable us to estimate reality by computing an estimated covariance matrix based on our theory (Σk). Fit compares the two covariance matrices. Guidelines for good fit provided in Chapter 9 apply. Here the researcher attempts to examine all aspects of construct validity through various empirical measures. The result is that CFA enables us to test the validity of a theoretical measurement model. CFA is quite different from EFA, which explores data to identify potential constructs. Many researchers conduct EFA on one or more separate samples before reaching the point of trying to confirm a model. EFA is an appropriate tool for identifying factors among multiple variables. As such, EFA results can be useful in helping to develop theory that will lead to a proposed measurement model. After a researcher builds confidence in a measurement model, CFA enters the picture.”

And in chapter 9 you'll find this:

„Cut-off Values for Fit Indices: The Magic .90, or Is that .95?

Although we know we need to complement the x2 with additional fit indices, one question still remains no matter what index is chosen: What is the appropriate cut-off value for that index? For most of the incremental fit statistics, accepting models producing values of .90 became standard practice in the early 1990s. However, some scholars concluded that .90 was too low and could lead to false conclusions, and by the end of the decade .95 had become the recommended standard for indices such as the TLI and CFI [25]. In general, .95 somehow became the magic number indicating good-fitting models, even though no empirical evidence supported such a development. Research has challenged the use of a single cut-off value for GOF indices, finding instead that a series of additional factors can affect the index values associated with acceptable fit. First, research using simulated data (for which the actual fit is known) provides counterarguments to these cut-off values and does not support .90 as a generally acceptable rule of thumb [25]. It demonstrates that at times even an incremental goodness-of-fit index above .90 would still be associated with a severely misspecified model. This suggests that cut-off values should be set higher than .90. Second, research continues to support the notion that model complexity unduly affects GOF indices, even with something as simple as just more indicators per construct [31]. Finally, the underlying distribution of data can influence fit indices [16]. In particular, as data become less appropriate for the particular estimation technique selected, the ability of fit indices to accurately reflect misspecification can vary. This issue seems to affect incremental fit indices more than absolute fit indices. What has become clear is that no single “magic” value always distinguishes good models from bad models. GOF must be interpreted in light of the characteristics of the research. It is interesting to compare these issues in SEM to the general lack of concern for establishing a magic R2  number in multiple regression. If a magic minimum R2 value of .5 had ever been imposed, it would be just an arbitrary limit that would exclude potentially meaningful research. So we should be cautious in adopting one size fits all standards. It is simply not practical to apply a single set of cut-off rules that apply for all SEM models of any type.”

For further reference, see the other 2 articles: Goretzko et al., 2023; McNeish & Wolf, 2023.

And in the book cited in reference 42, there is no clear criterion for CFI and TLI, but Byrne decides which is the better model based on the fit indices of competing models. In the case of AAQ-SF the opposite was the case, a model with weaker fit indices was adopted. Of course, I do not think that the validity of the psychometric model is more important than that of the psychological theory, but the acceptability of the fit indices is misleading and should not be used as a justification. It is acceptable to me that the fit indicator is lower, but it is more psychologically explainable (as you described below). I think that the study by Wolf (2023) can be a good help here.

References

Goretzko, D., Siemund, K., & Sterner, P. (2023). Evaluating Model Fit of Measurement Models in Confirmatory Factor Analysis. Educational and Psychological Measurement, 00131644231163813.

Hair, J.F.; Black, W.C.; Babin, B.; Anderson, R.E. SEM: Confirmatory factor analysis. In Multivariate Data Analysis, 8th ed. Cengage Learning: Andover, Hampshire, United Kingdom, 2019, pp. 658-698; ISBN 978-1-4737-5654-0.

McNeish, D., & Wolf, M. G. (2023). Dynamic fit index cutoffs for confirmatory factor analysis models. Psychological Methods, 28(1), 61.

Wolf, M. G. (2023). The problem with over-relying on quantitative evidence of validity. https://psyarxiv.com/v4nb2/download?format=pdf

We thank reviewer #3 for bringing this to our attention, and we agree and apologize for this oversight.

Accordingly, we now cite and showcase Chapter 9 in our reference list: Hair, J.F.; Black, W.C.; Babin, B.; Anderson, R.E. Structural equation modeling: An introduction. In Multivariate Data Analysis, 8th ed. Cengage Learning: Andover, Hampshire, United Kingdom, 2019, pp. 603-657; ISBN 978-1-4737-5654-0.

Based on this chapter, we have further amended Section 2.4 Statistical Analysis [paragraph 3] to the following: “Model Chi-Square (X2) statistics with p-values p > .05 are ideal; however, true-population models are prone to dismissal with samples of n ≥ 250 [41]. Adjunct goodness-of-fit assessments were: (1) Comparative Fit Index (CFI) and (2) Tucker-Lewis Index (TLI). Goodness-of-fit could be ascribed to models exhibiting CFI and TLI values of .94 or higher, with sample sizes > n = 250 and models espousing 12 observed variables or survey items [37]. The third assessment statistic is the Root-Mean-Square Error of Approximation (RMSEA), with values less than .07 [37] and a 90% confidence interval upper bound of up to .08 being reasonable approximations [38].”

In a similar vein, in the very last sentence of Section 2.4 Statistical Analysis [paragraph 3] we also now state: “The latter said ranges for goodness-of-fit assessments are more practical; researchers cannot judge the merits of their models using any “single magic value” (p. 642) [37]. A model’s validity should be judged using additional theoretical and practical criteria [38,44,46].”  

Please also note that in Section 2.4 Statistical Analysis [paragraph 3], we now refer to Byrne’s earlier work [38]. Byrne’s later work [42] is integrated into our paper later on and with respect to competing models in Section 4 Discussion [paragraph 5].

Other amendments corroborate the revised Section 2.4 Statistical Analysis [paragraph 3] and are as follows:

Section 3 Results [paragraph 3]: “In our categorical CFA of the proposed AAQ-SF structure [1], goodness-of-fit was evaluated using the CFI, the TLI, and the RMSEA and its 90% confidence interval [37,38,42,44] (Table 2). With the CFI and CFI (> .94), the results offer partial evidence of goodness-of-fit (X2 = 159.89, df = 51, p <.001; CFI = .980; TLI = .974). So too does the RMSEA statistic of .039, with its 90% confidence interval upper bound (.025-.052) being well below 0.08.”

Section 3 Results [paragraph 5]: “For this alternative two sub-domain PG structure (X2 = 116.14, df = 48, p <.001; X2 difference = 43.75, df difference = 3, p = .001; CFI = .927, TLI = .900; RMSEA = .05 [90% CI = .04-.065]), the CFI and TLI fall below our desired ranged [37].”

Section 3 Results [paragraph 7]: “In our convergence model (X2 = 454.12, df = 205, p <.001; CFI = .866, TLI = .849; RMSEA = .049 [90% CI = .043-.055]) (Table 4), the CFI and TLI fell below our desired range [37].”

In Section 4 Discussion [paragraph 8], we also refrain from qualitative claims about goodness of fit and now simply state “In our convergence model, the AAQ-SF subscales behaved as expected in relation to self-confidence and self-deprecation [38,44,46].”

2. The EFA could have been of help in arranging the items of the Psychological Growth scale (e.g. communality). The reliability of the 4-item scale is low, and the 2-2 item pairs are not really scales.

We thank you for this feedback. To conduct an exploratory analysis of the structure of the AAQ-SF, we would need to do so with one half of our studied sample and proceed in a confirmatory manner with the other half of our already studied sample. One is best poised for exploratory to confirmatory work using two different and thus wholly independent samples [98]. Laidlaw et al.’s (2017) AAQ-SF confirmatory analyses concluded with a third wholly independent studied sample of reputable stature (n = 792 persons) to not draw conclusions about the merits of the AAQ-SF with the same studied sample. We lack an independent validation sample. Others argue random sampling from a single studied sample does not guarantee equivalent samples for foraging exploratory and nested comparative work [99]. Rather, researchers should have participants from different sources mixed to compose a single larger sample split thereafter to best ensure the result depends on the true population factor model [99].

We therefore point out in Section 5 Limitations and Conclusion: “Further evidence is needed to confirm its validity with two independently studied international samples [1,98,99], and multiple goodness-of-fit indices, prevailing theories, and empirical literature at hand [1,38,44,46].

We also added to our reference list:

98.Henson, R.; Roberts, K.J. Use of exploratory factor analysis in published research – Common errors and some comment on improved practice. Educ. Psychol. Measurement 2006, 66, 393.https://doi.org/10.1177/0013164405282485

99.Lorenzo-Saver, U. SOLOMON: a method for splitting a sample into equivalent subsamples in factor analysis. Behav. Res. Meth. 2022, 54, 2665-2677. https://doi.org/10.3758/s13428-021-01750-y

We agree that the 2-2 item pairs are too austere to be formally deemed scales in their own right.

Accordingly, in Section 4 Discussion [paragraph 5], we now state: “Fit indices alone tell us that Laidlaw et al.’s [1] 4-item PG subscale best represents our sample data [37,42]. Goodness-of-fit indices are however but one yardstick with which to judge the validity of models [38,44,46]. There are theoretical and empirical reasons for further exploring the merits of PG as self-growth and generativity. Midlife adults are at the crossroads of youth and old age [11], navigating between adapting to age-related changes in functioning and investing in the well-being of younger and older generations through mentoring and caregiving [10]. Negative life events like physical health problems, divorce, and losses of paid work and loved ones begin to surface in people’s 40s and increase in their 50s [19]. Perhaps this is why self-compassion is so pivotal to positive mental health among people in their 40s and 50s [58]. There also appears to be a remarkable growth in wisdom about life’s uncertainties, including death and serious illness, between ages 51 and 99 [59]. It is important to focus on the ‘why’ of goodness-of-fit such as through qualitative explorations of midlife adults’ perceived meanings and relevance of all four PG items is important [60]. Qualifying (as opposed to just quantifying) observed austerities in goodness-of-fit can help us to better understand why items behave as they do across models and potential others to consider.”

Finally, in Section 5. Limitations and Conclusion, we add: “The present study findings about adults in their 50s support the 4-item PC and PL structure of the AAQ-SF and qualitative inquiry to better understand the meanings and relevance of items within the PG subscale [60].”

Please note that reference 60 pertains to: Wolf, M.G. The problem with over-relying on quantitative evidence of validity. Dissertation Chapter, University of California, Santa Barbara, December 8, 2022. Available online: https://psyarxiv.com/v4nb2/download?format=pdf (accessed September 25, 2023).